# Functional Properties of a Pitch-Based Carbon Fiber–Mortar Composite as a Thin Overlay for Concrete Pavement

**DOI:** 10.3390/ma12172753

**Published:** 2019-08-27

**Authors:** Jun Seok Lee, Inkyu Rhee

**Affiliations:** 1Institute of Bio-Housing, Chonnam National University, Gwangju 61186, Korea; 2Department of Civil Engineering, Chonnam National University, Gwangju 61186, Korea

**Keywords:** pitch-based carbon fiber, thermal conductivity, freeze-and-thaw resistance, bond strength

## Abstract

This experimental study investigated the utility of a pitch-based carbon fiber–mortar composite, which could replace polyacrylonitrile carbon fiber, as a thin overlay for concrete pavement. The objective was to explore the utility of the low-cost carbon fiber, which was produced via a melt-blown method, i.e., blowing at high pressure after melting the pitch residue following crude oil purification. The mechanical properties, durability, and thermal properties of the pitch-based carbon fiber were explored to maximize strength, durability, functionality, and economy by using micro-sized fibers that are closer in size to the constituents of cementitious materials. Melt-blown pitch-based carbon fiber has low individual fiber strength but generally excellent thermal conductivity. Thermal conductivity tests were conducted on mortar panels (560 mm × 560 mm; thickness = 25, 40 or 60 mm) containing 0, 0.4, 0.5 or 0.6 wt % pitch-based carbon fiber. The absolute thermal conductivity tended to improve with higher wt % of pitch-based carbon fiber, in the range of 9~11 W/°C. However, thermal conductivity tended to be lower under the 0.6 wt % condition, possibly due to the effect of dispersion. Compressive strength degradation was tested over 350 cycles of freezing and thawing: the strength of the 0.4, 0.5 or 0.6 wt % samples was 91, 89, and 82%, respectively, relative to the control specimen (0 wt %). Thus, all specimens had a compressive strength of 80% or more after 350 cycles compared to the control specimen. To test the adhesion performance for new thin overlays and old concrete surfaces, concrete cylinders (100 × 200 mm; thickness = 10 mm) were cut at an angle of 46 degrees, and the pitch-based carbon fiber-mortar composite was used to bond the various sections. The bond strength of the test specimens was more than twice that of the reference specimen.

## 1. Introduction

Sustainable construction materials, such as cement composite, must have high structural strength and multi-functionality. It is vital to appropriately maintain the function, and thus ensure the safety, of existing infrastructure, and to use a durable concrete that can be rapidly assessed in terms of structure, function, and safety. Researchers are currently exploring various methods for monitoring the structural health of the civil infrastructure. Conductive materials, including carbon-based materials (carbon fibers [1,2,3,4,5,6], carbon nanofibers [7,8,9,10,11], carbon nanofibers/nanotubes [12,13,14,15], and graphenes [16]), are now being included in non-conductive concrete to improve the functionality of sensors therein [17,18,19]. This could lead to changes in the basic mechanical properties (e.g., compressive and tensile strength) and durability (e.g., abrasion resistance, freeze-thaw-performance, permeability) of the concrete [20,21,22,23,24,25,26,27,28]. Carbon nanotubes and graphene are nanomaterials with particle sizes on the order of 10^−9^ m; the volume size of mortar and concrete is 0.05~0.1 m, so the size difference is in the range of 10^7^–10^8^-fold. This difference might be an important consideration when developing cement composites. Of course, it could be dependent on the properties of the inclusions of the composite, one of which is the granulation of both cement and aggregate. Additionally, the relationship between nanomaterials and mortar or concrete, which is the hierarchical linking relationship between different scales [6,16], has a large impact on overall performance, in terms of both mechanical and multi-functional properties. In particular, mechanical strength may be low when there is a large difference between nanomaterial particle size and the volume size of mortar or concrete, and it is also relatively difficult to achieve homogeneous dispersion [29,30,31,32,33,34] with respect to electric conductivity. As mentioned in the above, the scale difference increase/decrease the homogenized dispersibility in a statistical sense. Let say, if the specimen size is relatively large, then average dispersibility over the whole domain of the specimen has a tendency to be lowered. We had the lower electrical conductivity at 0.6 wt % specimens with larger specimen thickness than 0.5 wt % specimen. This has been discussed in a later section. Minimizing the material size difference requires the inclusion of micro-sized, rather than nano-sized, materials in cement composites. In the present study, the size difference among the admixture, fine aggregate, and coarse aggregate was reduced to less than 10^1^~10^5^ folds (e.g., normal-weight concrete: 0.15 μm for silica fume, 10 μm for cement, 0.5~5 mm for sand, and >25 mm for gravels.), which is expected to considerably improve mechanical and functional performance, and promote ease of use. In previous construction applications, carbon fiber has been used to repair and strengthen reinforced concrete structures, such as buildings and bridges: in these applications, a polyacrylonitrile (PAN)-based carbon fiber fabric is attached to the exterior of the structure to be repaired and reinforced with an epoxy-based adhesive rather than a mixture of carbon fibers. Approximately, PAN series carbon fiber cost 10~15$ per kg [35] and less than a dollar for pitch-based fiber used in this study. This price for pitch-based fiber is not defined yet because it is the residue from the oil refinery plant (GS Caltex, Daejeon, Korea). However, anisotropic (mesophase) pitch-based fiber has a very high price and outstanding fiber performance for both mechanical and functional aspects [35,36]. This reinforcement technique can ensure effective strength without enlarging the cross-section of the structure. However, it is also expensive and has the disadvantage of frequent delamination failure, which separates the attachment surfaces; more rapid local adhesion occurs when structures are exposed to constant moisture (e.g., underwater structures). The federal governments and state departments of transportation in the United States and Canada are interested in using calcium chloride to maintain roads, where this can prevent freezing and thawing as well as chloride penetration [19]. Additionally, because snow melts more slowly on bridges than on the surrounding ground, care is needed during snow removal and de-icing [22,23,24,25,26,27,28]. The most widely used carbon fibers are manufactured using a PAN polymer as a precursor. As an alternative, short isotropic carbon fibers may be obtained using petroleum residue as a precursor via a melt-blown method [6,35,36]. The melt-blown method is less expensive and more ecofriendly: pitch-based carbon fiber is recycled from oil industry waste, and its use leads to a productivity 10-fold higher than that of the conventional melt spinning process. The compressive strength and electrical resistance of isotropic pitch-based carbon fibers in cementitious materials have been explored previously, to determine their feasibility for use as a functional construction material [6,37,38,39,40,41]. The present study explored the thermal conductivity, bond strength, and freeze-and-thaw resistance of cement paste using isotropic pitch-based short carbon fiber as a reinforcing filler. Slag and a combination of a viscosity agent and superplasticizer were explored as possible aids for dispersion in cement paste and mortar. The fiber length and wt % of the tailored carbon fiber were 5 mm, and 0, 0.4, 0.5 or 0.6 wt %, respectively. Cement mortar composite panels, cylinders, and cubes were cast to evaluate the thermal conductivity, bond strength, and freeze-and-thaw resistance of cement paste using isotropic pitch-based short carbon fiber as a reinforcing filler. Here, the focus was on pitch-based carbon fiber, which could replace PAN carbon fiber. Few previous studies have investigated pitch-based carbon fibers [6], and the purpose of the present study was to maximize the strength, durability, and functionality of concrete by using low-cost micro-sized fibers that are closer in size to the constituents of cementitious materials.

## 2. Materials and Methods

Mortar panels (560 × 560 mm; thickness = 25, 40 or 60 mm) containing pitch-based carbon fiber were fabricated for thermal conductivity tests, as illustrated in Figure 1a. In total, 12 mortar panel specimens were cast to investigate the thermal performance of pitch-based carbon fiber-mortar composites. The heating cable had a diameter of 7 mm, a single multi-conductor, polytetrafluoroethylene (PTFE) insulation, and additional outer layers (reinforced insulation layer, tinned copper braid, and outer jacket). The capacity of the cable was 35 W/m at 10 °C in air, for nominal power output; the maximum temperature was 85 °C (at 20 °C in the air). As shown in Figure 1a,c, 4 m of heating cable was arranged in a zigzag pattern with lateral spacing of 100 mm, and eight thermocouples were embedded or glued before and after casting the mortar panels. The heat at the heat source, bottom center of the heating cable, and the top surface (at five locations), as well as the ambient temperature, were measured to obtain time-dependent heat information, as illustrated in Figure 1. A data logger (Agilent Technologies, Santa Clara, CA, USA), current regulator, power meter, and timer were synchronized to acquire temperature, voltage, and resistance data for the test specimens and cable, as shown in Figure 1b. An infrared camera was used to capture the temperature contour on the top surface of the specimens, as shown in Figure 1d. The pitch-based carbon fiber manufactured by the melt-blown process used in this study is shown in Figure 1e. Table 1 lists the basic properties of this pitch-based carbon fiber.

The mortar panel specimens contained 0, 0.4, 0.5, or 0.6 wt % pitch-based carbon fibers (relative to the total weight of the cement). Admixtures of chemicals, such as slag, superplasticizer, and viscosity agents, were added to enhance the dispersion of pitch-based carbon fiber inside the mortar specimens (Table 2). Cube-shaped specimens were subjected to 350 cycles of freezing and thawing. Specimens were labeled as PT25, C04T25, C05T25, C06T25, etc. according to the carbon fiber weight percentage and panel thickness. Mixing was conducted via the following procedure: (1) dry mix with cement, sand, and slag for 10 min, (2) wet mix for 7 min with two-thirds of the required water and pitch-based carbon fiber, (3) wet mix for another 5 min with superplasticizer and viscosity agents, and (4) stabilize for 3 min before casting. Specimens were demolded 1 day after casting and placed into a water chamber for 27 days for curing. Ordinary Portland cement (type 1) and slag from the local area (Gwangju, Korea) was used. The mixing process was performed using a standard Hobart-like mixer (Kenwood mixer), as specified in ASTM (American Society for Testing and Materials) C 305-14. A polycarboxylic acid-based admixture was applied to enhance the flow of the carbon fiber/cement composites and to facilitate dispersion of the fiber cluster.

All surfaces (except one side in the thickness direction) of the thermal test specimens were insulated using expanded polystyrene (EPS) panel to allow the inlet and outlet of the heating cable to enter at the bottom of the panel, as shown in Figure 1d. Initially, the heat at the center of the cable was controlled at 70 °C. The duration of the heating process was 6 h, and the temperature, power, and resistance were sampled every 30 s. The average power used for heating was 90 ± 6.26 W, and the average resistance was stable at 2.23 ± 0.2 ohm. We attempted to quantify changes in thermal conductivity in the mortar specimens in terms of the temperature gradient between the heat source at the bottom and the top surface, according to the power used. The temperature gradient was verified by a simple 1D transient heat transfer analysis. The in-house freeze-and-thaw experiments were carried out in the range of temperature from −24 to 15 °C under 350 cycles of environmental fatigue loading. The bond strength of pitch-based carbon fiber (CF) mortar composite is an important thin overlay as a thermally conductive mortar on the top of existing concrete pavement. After modifying the ASTM C-882 test method, a 10 mm pitch-based CF mortar thin overlay was attached to a 100 × 200 mm concrete cylinder having the slanted cross-section.

## 3. Results

First, thermal conductivity tests were conducted without top surface insulation, as shown in Figure 2. However, this analysis can easily be affected by fluctuations in ambient temperature. Therefore, to measure the thermal conductivity of the panels, their top surfaces were insulated using an EPS panel. Thermal conductivity *k* is the ability of a material to transmit heat; it is measured in watts per square meter of surface area for a temperature gradient of 1 °C per unit thickness of 1 m. Similarly, absolute thermal conductivity can be expressed as *k_a_*, defined by Equation (1). Table 3 and Figure 3 present the variations in absolute thermal conductivity according to the wt % and thickness of the composite panels. Absolute thermal conductivity was enhanced by increasing the wt % of the pitch-based carbon fiber and tended to decrease along relatively thick panels. *T*_0_ is the bottom temperature of the panel, *T*_1_, *T*_2_, *T*_3_, and *T*_5_ are the temperatures between the edge and the center of the top of the panel, and *T*_4_ is the temperature at the center of the top of the panel.

(1)$$k=\frac{P}{\left(A\cdot \Delta T/t\right)},k_{a}=\frac{P}{\Delta T}$$

The 1D time-dependent heat conduction analysis was used to calibrate the heat in terms of the specific heat, convection coefficient, and conductivity of the carbon fiber-motar composite. The basic differential equation for time-dependent heat conduction [42] is as follows:(2)$$K\cdot T+M\cdot \dot{T}=F$$
where **K**, **M**, and **F** are the conductivity, mass matrix, and heat source vector, respectively, defined in global coordinates, and **T** and $\dot{T}$
are the temperature and its gradient vector, respectively. Elemental conductivity, mass, and heat sources can be expressed in simple 1D space using Equation (3), where *k_xx_* is the conductivity in W/m °C, A is the area of the composite panel (0.56 × 0.56 m^2^), *L_e_* is the element length in the thickness direction of the panel, *c* is the heat capacity, 1000 J/kg·°C, *ρ* is the density of the composite panel, 2110 kg/m^3^*, h* is the convection coefficient, 12~18 W/m^2^ °C, *P* is the perimeter of the heat convection area, *m*, and T_∞_ is the ambient temperature in the laboratory.

(3)$$k=\frac{k_{xx}A}{L_{e}}\left[\begin{array}{cc} 1 & -1\\ -1 & 1 \end{array}\right],\;m=\frac{c\rho AL_{e}}{6}\left[\begin{array}{cc} 2 & 1\\ 1 & 2 \end{array}\right],\;f=\frac{hT_{\infty }PL_{e}}{2}\left[\begin{array}{c} 1\\ 1 \end{array}\right]$$

For numerical time integration, the time-dependent temperature distribution can be solved based on Equation (2). Equation (4) assumes that two temperature states, **T***_i_* at time t*_i_* and **T***_i+_*_1_ at time t*_i+_*_1_, are related by:(4)$$\dot{T}=T_{i+1}-T_{i}=\left[\left(1-\beta \right){\dot{T}}_{i}+\beta {\dot{T}}_{i+1}\right]\Delta t$$

Plugging Equation (4) into Equation (2) yields Equation (5):(5)$$K\cdot \left[\left(1-\beta \right)T_{i}+\beta T_{i+1}\right]+M\cdot \left[\left(1-\beta \right){\dot{T}}_{i}+\beta {\dot{T}}_{i+1}\right]=\left(1-\beta \right)F_{i}+\beta F_{i+1}$$

Equation (6) can be rearranged by eliminating the time derivative terms:(6)$$\left(\frac{1}{\Delta t}M+\beta K\right)T_{i+1}=\left[\frac{1}{\Delta t}M-\left(1-\beta \right)K\right]T_{i}+\left(1-\beta \right)F_{i}+\beta F_{i+1}$$

Depending on the value of *β*, the time step Δ*t* may have an upper limit above which the numerical analysis is unstable. If *β* < 1/2, the largest Δ*t* for an unconditionally stable condition would be:(7)$$\Delta t=\frac{2}{\left(1-2\beta \right)\lambda_{\max}},\;\left(K-\lambda M\right)\cdot T=0$$
where $\lambda_{\max}$ is the largest eigenvalue, and *β* = 2/3 based on the Crank–Nicolson method, which is unconditionally stable in Equation (7). This 1D transient heat transfer analysis was performed in the Matlab environment (MathWorks, Natick, MA, USA). The purpose of the analysis was to assess thermal conductivity experimentally and numerically. Figure 4 presents the results regarding transient heat transfer through panel C0.6T25 (0.6 wt %; thickness = 25 mm), and Table 4 presents the results regarding the conductivity and convective coefficient, *k_Matlab_*, and *h_Matlab_* respectively from the analyses. Because a narrow convective boundary surface (0.56 m × Thk.: transverse section where the heating cable is in and out from the specimen) was used to allow the inlet and outlet of the heating cable to enter at the bottom of the panel, the change in convective coefficient *h* must be limited. There was good agreement between the experimental and numerical results with respect to the change in coefficient *h*. The heat capacity and density of the composite panel were kept constant at *c =* 1000 J/kg·°C and ρ = 2110 kg/m^3^, respectively. The heat capacity *c* value would not alter much if the temperature gradient is not abruptly changed, especially under a normal temperature range, such as 100 °C. The typical heat capacity would be in the range of 880~1000 J/kg·°C [43]. However, this would be altered with a rapid rise of temperature, such as a fire. The density of the specimen was set to 2110 kg/m^3^, and this averaged value was measured by the authors.

Cube-shaped mortar specimens were prepared for freeze-thaw resistance tests. In-house freeze-thaw experiments (350 cycles) were carried out at temperatures ranging from −24 to 15 °C, as shown in Figure 5a. Static compressive strength, as shown in Figure 5b, was degraded by 9.4~18.1%. The strength was best preserved under the 0.5 wt % condition, which also yielded the best heat conduction performance. The change of specimen weight was recorded, and weight loss could be seen by the fatigue loading in Figure 5c. The specimen size for freeze-and-thaw resistance was 50 mm cube. Normally, 200~300 cycles were used for evaluating this resistance capacity [43]. Due to the malfunction of the in-house machine (controller), 350 cycles were done accidentally. Air-dried specimens were the control specimen with different wt % of fibers without freeze-and-thaw fatigue condition. The strength reduction by increasing fiber wt % should be eliminated from the overall reduction after 350 cycles of freeze-and-thaw fatigue loading. The bond strength of pitch-based carbon fiber-mortar composite, for use as a thin overlay of thermally conductive mortar on top of existing concrete pavement, is an important factor. As a modification of ASTM C-882 (Standard Test Method for Bond Strength of Epoxy-Resin Systems Used with Concrete; American Society for Testing and Materials, USA) (Figure 6), a 10 mm pitch-based carbon fiber-mortar was overlaid on a 100 × 200 mm concrete cylinder with a slanted cross-section. During the compressive strength test, bond strength was measured according to the extent of the area of bond failure and the breakage load; bonding was strongest under the 0.5 wt % condition. Overall, 0.5 wt % of pitch-based carbon fiber yielded the best compressive and adhesion strength, electrical and thermal conductivity, and freeze-thaw performance.

## 4. Discussion and Conclusions

This study assessed the utility of a low-cost carbon fiber-mortar composite, produced using a melt-blown method, i.e., blowing at high pressure after melting the pitch residue following crude oil purification, as a thin overlay for concrete pavement. The melt-blown method is less expensive and more ecofriendly: pitch-based carbon fiber is recycled from oil industry waste, and its affordable price and mass production may lead to increase the feasibility of massive use on the civil engineering applications. However, the basic mechanical properties of this pitch-based carbon fiber, as discussed earlier in this paper, has a quite small contribution on the cement-CF composite action. Thus, we seek the functionality of pitch-based fiber to take advantage of its better thermal conductivity. This was achieved by testing a thin overlay specimen with cement-pitch-based carbon fiber composite for de-icing the concrete pavement. To this end, three main topics were selected: (1) thermal conductivity for de-icing efficiency, (2) freeze-thaw resistance against harsh environmental deterioration, and (3) adhesion performance for bonding capability between old and new layers.

The conclusions are summarized below:(1)Thermal conductivity: The melt-blown pitch-based carbon fiber was characterized by low-strength individual fibers, but showed generally excellent thermal conductivity in tests of various mortar panels (0, 0.4, 0.5 or 0.6 wt % pitch-based carbon fiber content relative to the total weight of the cement, and thickness of 25, 40 or 60 mm). The absolute thermal conductivity tended to improve with a higher wt % of pitch-based carbon fiber in the range of 9~11 W/°C. However, the thermal conductivity tended to be lower under the 0.6 wt % condition, possibly due to the effect of dispersion.(2)Freeze-thaw resistance: Compressive strength degradation of 50 mm cube-shaped mortar specimens following 350 freeze-thaw cycles were measured at 91%, 89%, and 82% for the 0.4, 0.5, and 0.6 wt % samples, respectively, relative to the control specimen (0 wt %). Thus, all specimens had a strength of 80% or more after 350 cycles relative to the control specimen.(3)Adhesion performance: the adhesion performance of new and old concrete surfaces was tested. Concrete cylinders (100 mm × 200 mm; thickness = 10 mm) were cut at an angle of 46 degrees, and the pitch-based carbon fiber-mortar composite was used to bond the various sections. The bond strength of the test specimens was more than twice that of the reference specimen, although, at 0.6 wt %, the bond strength began to decline.

## Figures and Tables

**Figure 1 materials-12-02753-f001:**
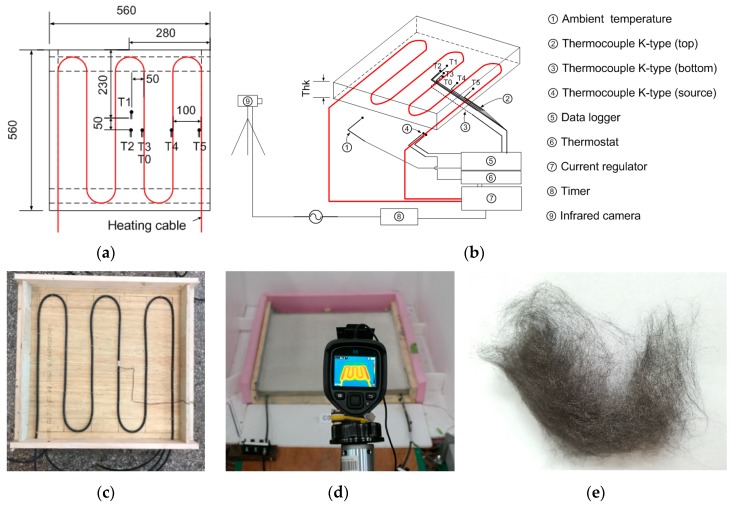
Experimental setup for the thermal conductivity measurements: (**a**) mortar panel dimensions and heat coil arrangement, (**b**) heat measurements obtained by the thermocouples and infrared camera, (**c**) wooden mold used for the panel specimens, (**d**) infrared camera used for obtaining surface measurements, and (**e**) pitch-based carbon fiber (GS Caltex, Daejeon, Korea).

**Figure 2 materials-12-02753-f002:**
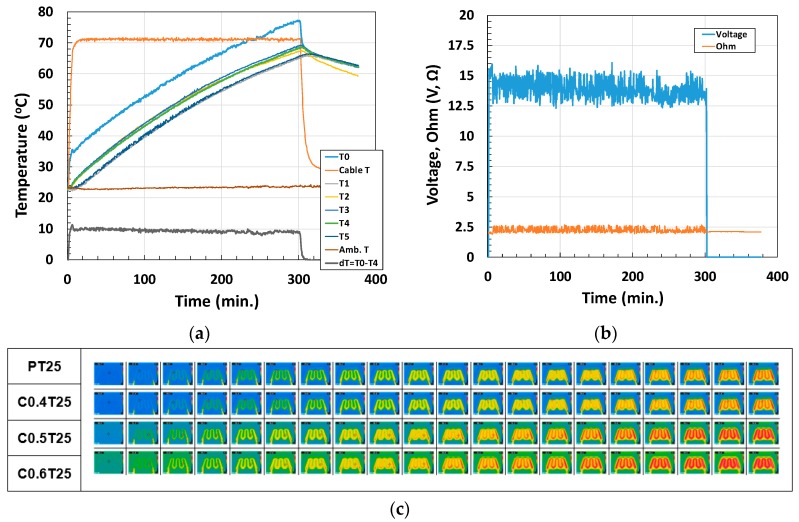
Changes in temperature: (**a**) time-dependent temperature at *T*_0_–*T*_4_, and at the heat source (cable temperature), and the ambient temperature and temperature gradient, (**b**) voltage and resistance of the cable while heat was applied, and (**c**) temperature distribution on the top surface measured using an infrared camera (every 20 min over the 6 h heating process). C0.4, C0.5, and C0.6: mortar composite with 0.4 wt %, 0.5 wt %, and 0.6 wt % of pitch-based carbon fiber, T25: 25 mm specimen thickness.

**Figure 3 materials-12-02753-f003:**
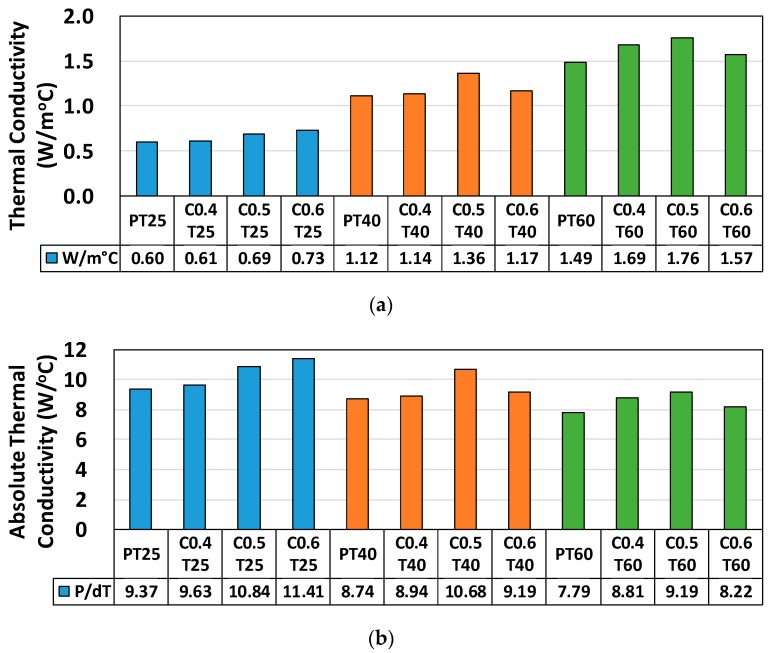
Measured properties: (**a**) thermal conductivity (W/m °C), (**b**) absolute thermal conductivity (W/°C), (**c**) temperature gradient, Δ*T* = *T*_0_ − *T*_4_, and (**d**) power (watts) in use for the heating process.

**Figure 4 materials-12-02753-f004:**
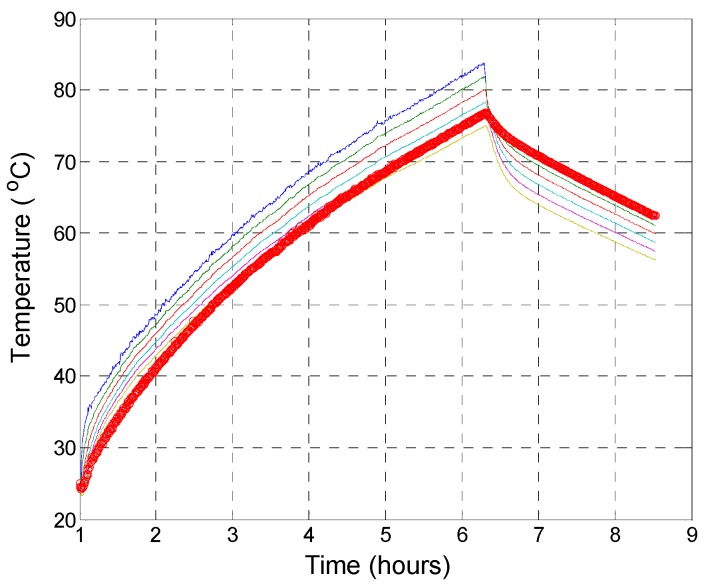
Transient heat analysis of the top surface of specimen C0.6T25: *k*_c0.6T25_ = 0.728 W/m °C, *h*_c0.6T25_ = 8 W/m^2^°C, *c*_c0.6T25_ = 1000 J/kg·°C, ρ = 2110 kg/m^3^ (thick red line).

**Figure 5 materials-12-02753-f005:**
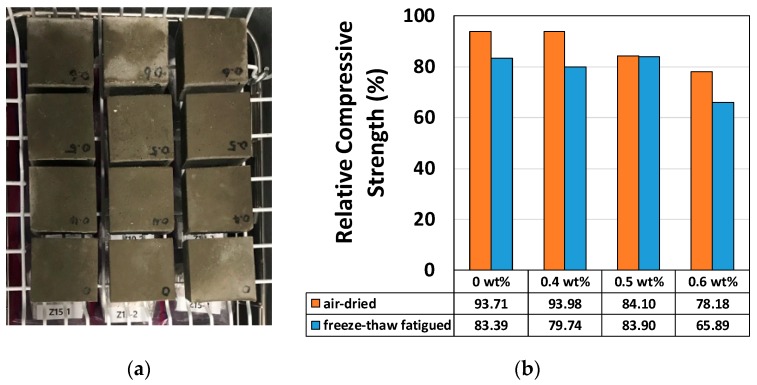
Freeze-thaw resistance of pitch-based carbon fiber-mortar composite: (**a**) 350 freeze-thaw cycles were applied at 50 mm× 50 mm × 50 mm (nine specimens with pitch-based carbon fiber content of 0–0.6 wt %), (**b**) relative strength relative to reference specimen strength (%) and (**c**) specimen weight (g): fatigued vs. air-dried specimen.

**Figure 6 materials-12-02753-f006:**
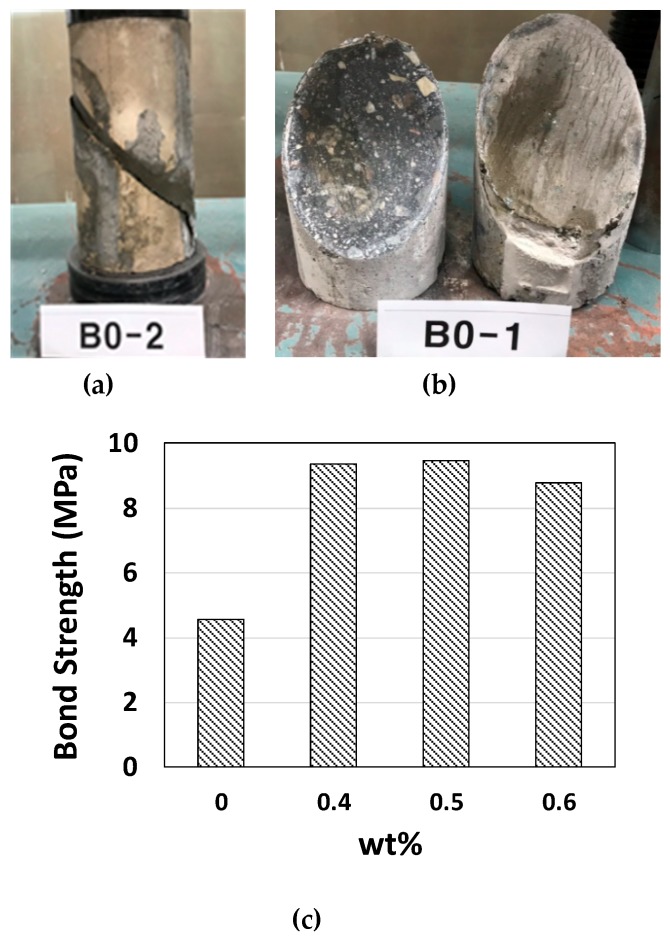
(**a**) Bond strength of nine concrete specimens (100 × 200 mm, pitch-based carbon fiber content of 0–0.6 wt %) thin slanted gap of 10 mm, (**b**) photograph of the specimen subjected to bond strength testing, and (**c**) bond strength by mixing ratio.

**Table 1 materials-12-02753-t001:** Basic properties of the pitch- and PAN (polyacrylonitrile)-based carbon fibers [6].

Physical/Mechanical Properties	Pitch-Based Carbon Fiber (GS-Caltex, Daejeon, Korea)	PAN-Based Carbon Fiber (T-300; Toray,Tokyo, Japan)
Fiber diameter	11.96 ± 5.24 μm	7.11 ± 2 μm
Tensile strength	0.35 ± 0.13 GPa	3.55 ± 0.56 GPa
Tensile modulus	25.89 ± 5.68 GPa	228.52 ± 7.67 GPa
Elongation at break	1.5 ± 0.58%	1.69 ± 0.23%
Electrical conductivity	2.2 × 10^2^ S/cm	5.88 × 10^2^ S/cm
Specific gravity	1.543 g/cm^3^	1.76 g/cm^3^

**Table 2 materials-12-02753-t002:** Mix design for pitch-based carbon fiber-mortar composites.

Specimen		CF (g)	C (g)	W (g)	S (g)	SL (g)	SP (mL)	VA (g)	Geometry
P		0	720	360	576	216	2.8	0.4	Cube (6 each)
C0.4		2.9							
C0.5		3.6							
C0.6		4.3							
P	T25	0	7526	3763	6021	2258	28.9	3.8	Panel (12 each)
C0.4		30.1							
C0.5		37.6							
C0.6		45.2							
P	T40	0	12,042	6021	9634	3613	46.2	6	
C0.4		48.2							
C0.5		60.2							
C0.6		72.3							
P	T60	0	18,063	9032	14,451	5419	69.2	9	
C0.4		72.3							
C0.5		90.3							
C0.6		108.4							

CF: pitch-based carbon fiber, C: cement, W: water, S: sands, SL: slag, SP: superplasticizer, VA: viscosity agent, P: plain mortar w/o CF, C0.4, C0.5, and C0.6: mortar composite w/0.4 wt %, 0.5 wt %, and 0.6 wt % of CF, T25, T40, and T60: 25 mm, 40 mm, and 60 mm specimen thickness.

**Table 3 materials-12-02753-t003:** Thermal conductivities of test specimens by average power and temperature gradient.

Specimen	*P* (W)	Δ*T* (°C)	*k* (W/m °C)	*k_a_* (W/°C)
PT25	87.2	9.3	0.597	9.372
PT40	89.6	10.3	1.115	8.744
PT60	82.4	10.6	1.490	7.790
C0.4T25	84.2	8.7	0.614	9.634
C0.4T40	85.1	9.5	1.140	8.941
C0.4T60	94.1	10.7	1.686	8.812
C0.5T25	91.8	8.5	0.692	10.843
C0.5T40	91.7	8.6	1.363	10.683
C0.5T60	99.9	10.9	1.758	9.186
C0.6T25	82.5	7.2	0.728	11.408
C0.6T40	100.8	11	1.172	9.190
C0.6T60	93.8	11.4	1.573	8.222

P: power in use, ΔT: temperature gradient (*T*_0_–*T*_4_), ***k***: thermal conductivity in W/m °C, and ***k_a_***: absolute thermal conductivity in W/°C.

**Table 4 materials-12-02753-t004:** Thermal coefficients under constant heat capacity and density conditions *(c =* 1000 J/kg·°C and ρ = 2110 kg/m^3^, respectively).

Specimen	*k* (W/m °C)	*k_Matlab_* (W/m °C)	*h_Matlab_* (W/m^2^ °C)
PT25	0.597	0.560	18
PT40	1.115	1.120	18
PT60	1.490	1.700	18
C0.4T25	0.614	0.800	12
C0.4T40	1.140	1.140	12
C0.4T60	1.686	1.620	12
C0.5T25	0.692	0.800	12
C0.5T40	1.363	1.363	12
C0.5T60	1.758	1.758	12
C0.6T25	0.728	0.728	8
C0.6T40	1.172	1.100	12
C0.6T60	1.573	1.573	12

***k***: thermal conductivity from experiments, ***k_Matlab_***, and ***h_Matlab_***: thermal conductivity and convection coefficient estimated by 1D transient heat analysis using Matlab.

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
