# Peer review of "Functional Properties of a Pitch-Based Carbon Fiber–Mortar Composite as a Thin Overlay for Concrete Pavement"

_materials, 2019, doi:10.3390/ma12172753_

Round 1

Reviewer 1 Report

1. The word “infrastructure” is repeated in line 34.

2. Is there standards or references about the measurement of thermal performance of pitch-based CF and mortar composites?

3. In Table 1, “S/cm” is the unit of the electrical conductivity instead of electrical resistivity.

4. What is the meaning of “P” in Table 2, and this letter also appears later. Please annotate abbreviations to further clarify the difference.

5. It is easy to understand that the thermal conductivity increase with adding pitch based carbon fibers, what is the reason that thermal conductivity decrease when the content of carbon fibers is 0.6%. Is there any evidence about the effect of dispersion.

6. Why does the bond strength increase after adding carbon fibers then decrease when the content of carbon fibers is 0.6%?

7. Is the paper to be published in Materials?Why the article template is Coatings

Author Response

The authors appreciate greatly for the valuable comments of the reviewer on the manuscript. We updated the manuscript based on the comments and suggestions from the reviewer. The responses from the authors are as follows:

Reviewer #1 :

(1) The word “infrastructure” is repeated in line 34.

Yes, the repeated word was deleted.

(2) Is there standards or references about the measurement of thermal performance of pitch-based CF and mortar composites?

Unfortunately, there is no standard regarding on this. However, the authors followed the standard thermal conductivity measure technique [43].

(3) In Table 1, “S/cm” is the unit of the electrical conductivity instead of electrical resistivity.

The term, resistivity has been updated.

(4) What is the meaning of “P” in Table 2, and this letter also appears later. Please annotate abbreviations to further clarify the difference.

"P" indicates the control specimen without fiber inclusion and noted in Table 2.

(5) It is easy to understand that the thermal conductivity increase with adding pitch based carbon fibers, what is the reason that thermal conductivity decrease when the content of carbon fibers is 0.6%. Is there any evidence about the effect of dispersion.

Unfortunately, we could not explore the dispersibility on each different wt% specimen, see if there were good or poor distribution of fiber inclusion (e.g. section analysis, etc). However, statistically speaking, a larger concentration and relatively larger size of specimen may leads decrease of dispersibility.

(6) Why does the bond strength increase after adding carbon fibers then decrease when the content of carbon fibers is 0.6%?

We believe that the higher concentration of fiber dose not guarantee the increasing tendency, especially for the mechanical performance such as compressive strength, bond strength etc. There may be the optimal wt% for the specific matrix material which is mortar in our case. This wt% would be approximately 0.5 wt%.

(7) Is the paper to be published in Materials?Why the article template is Coatings

The authors originally submitted the manuscript without any specific stationary form. The authors presume this was from the Journal office. We will query to the Journal office regarding on this later on.

Reviewer 2 Report

The study deals with the influence of pitch-based carbon fibers content on various properties of cement mortars. Several parts should be revised before publishing the manuscript in the journal of Materials.

The specific comments are as follows:

Please improve quality of graphs. For instance, text sizes in Figures 1, 3, 5 and 10 are too small.

The information about the type of cement is missing in the article. Please add it.

The structure of the article should be revised and changed into: Introduction, Materials and Methods, Results, Discussion and Conclusions. As for now, the information about the materials that were used by the Authors in the study, except fibers, is missing. Results and methods are presented simultaneously. There is no discussion section and conclusions could be formed more precisely if the results of the tests conducted by the Authors were confronted with other research.   

Please rewrite the Abstract and include specific statements about your research.

[Line 39] 32 references in one place?

[Line 42-48] what is representative volume size of a mortar? Tests of various properties of cement composites are performed on elements of the size or bigger that Authors mentioned in the text, however those properties depend on the properties of the ingredients of the composite, one of which is the granulation of both cement and aggregate.

'the systematic linking relationship [...]' - between what?

'[...] the portion that contributes to strength [...] - portion of what? If Authors want to describe those phenomena, specific values and references need to me mentioned in text. why is it difficult to secure homogeneous dispersability in the case of electric conductivity? Please elaborate.

[Line 51] What about cement? Particle sizes of that ingredient are measured in micro scale.

[Line 55] Please add the approx price value and reference.

[Line 75-76] If Authors want to discuss results in the introduction, presenting precise results would be necessary.

[Table 2] Were Authors adding silica fume (Line 75) or slag (Table 2) to the composite? Please explain.

[Line 203] How do Authors know if those values are true? If tests regarding c and density were conducted please indicate it in the text.

[Line 211-217] What was the size of mortar specimens for freeze-thaw resistance? Why Authors decided to run 350 cycles? Why were the specimens not fatigued by freeze-thaw cycles air-dried? What code or technical paper were Authors following while performing those tests? 

[Fig 5c] Please make the chart bigger by deleting 5b. What were the actual compressive strength results for each of the series? Have Authors tested the change in mass of the specimens?

Conclusion section needs to be rewritten and provide specific conclusions that can be made o the basis of the research Authors have performed.

Author Response

The authors appreciate greatly for the valuable comments of the reviewer on the manuscript. We updated the manuscript based on the comments and suggestions from the reviewer. Please find the attachment for the responses from the authors.

Reviewer 3 Report

The technical content in this paper is strong and the work presented is very relevant.

However, there are numerous sentences throughout the paper that don't make much sense as they are incorrect or poorly written. I have highlighted some of these in the attachment.

There are several graphs that have not been discussed.

Author Response

The authors appreciate greatly for the valuable comments of the reviewer on the manuscript. We updated the manuscript based on the comments and suggestions from the reviewer. The responses from the authors are as follows:

(1) However, there are numerous sentences throughout the paper that don't make much sense as they are incorrect or poorly written. I have highlighted some of these in the attachment.

Thank you for your comment. The manuscript is now edited by native English speaker.

The English in this document has been checked by at least two professional editors, both native speakers of English. For a certificate, please see:

http://www.textcheck.com/certificate/bAR5dp

(2) There are several graphs that have not been discussed.

Yes, the explanation for graphs has been commented accordingly. Please refer the revised manuscript.